# Diet and Lipid-Lowering Nutraceuticals in Pediatric Patients with Familial Hypercholesterolemia

**DOI:** 10.3390/children11020250

**Published:** 2024-02-15

**Authors:** Maria Elena Capra, Giacomo Biasucci, Giuseppe Banderali, Andrea Vania, Cristina Pederiva

**Affiliations:** 1Centre for Pediatric Dyslipidemias, Pediatrics and Neonatology Unit, University of Parma, Guglielmo da Saliceto Hospital, 29121 Piacenza, Italy; m.capra@ausl.pc.it (M.E.C.); g.biasucci@ausl.pc.it (G.B.); 2Department of Translational Medical and Surgical Sciences, University of Parma, 43126 Parma, Italy; 3Department of Medicine and Surgery, University of Parma, 43126 Parma, Italy; 4Clinical Service for Dyslipidemias, Study and Prevention of Atherosclerosis in Childhood, Pediatrics Unit, ASST-Santi Paolo e Carlo, 20142 Milan, Italy; giuseppe.banderali@asst-santipaolocarlo.it (G.B.); cristina.pederiva@asst-santipaolocarlo.it (C.P.); 5Independent Researcher, Member of SINUPE (Italian Society of Pediatric Nutrition) Directory Board, 00162 Rome, Italy

**Keywords:** nutritional intervention, diet, nutraceuticals, dyslipidemia, cardiovascular disease, children

## Abstract

Familial hypercholesterolemia is a genetically determined disease characterized by elevated plasma total and LDL cholesterol levels from the very first years of life, leading to early atherosclerosis. Nutritional intervention is the first-line treatment, complemented with nutraceuticals and drug therapy when necessary. Nutraceuticals with a lipid-lowering effect have been extensively studied in the past few decades, and have been recently included in international guidelines as a complement to nutritional and pharmacological treatment in subjects with dyslipidemia. In this review, we explore current nutritional interventions for dyslipidemia in childhood, with a specific focus on the main nutraceuticals studied for treating severe dyslipidemia in pediatric patients. Additionally, we briefly describe their primary mechanisms of action and highlight the advantages and risks associated with the use of lipid-lowering nutraceuticals in childhood.

## 1. Introduction

### 1.1. Early Atherosclerosis

Coronary heart disease (CHD) related to atherosclerosis can be considered as the primary determinant of death and diseases worldwide, particularly in high-income countries. In the European region, deaths in subjects older than 18 years are caused by CHD in 45% of cases, with a prevalence of 49% in women and 40% in men [1]. It is universally acknowledged that atherosclerosis begins even before birth, in the fetal period, and progresses throughout one’s lifetime, from birth to adult age. The increasing aggregate risk of the exposure to elevated Low-Density Lipoprotein (LDL) cholesterol plasma values accelerates the atherosclerosis pathway. Prolonged exposure to LDL cholesterol results in an increased risk of CHD. In subjects with homozygous familial hypercholesterolemia (HoFH), LDL cholesterol levels are markedly high from the first days of life. In these subjects, myocardial infarction and angina pectoris have been reported as early as the first decade of life. The earlier that pediatric subjects at high CHD risk are diagnosed and treated, the more favorable their clinical outcomes will be. Taking prompt action on nutritional habits, lifestyle and, if necessary, initiating drug treatment from childhood can effectively alter the natural course of the disease. This intervention can potentially “add decades of life”, resulting in a significant reduction in coronary heart disease during adulthood. 

### 1.2. Early Treatment

As asserted by the principal Guidelines and Consensus Statements from the American Academy of Pediatrics (AAP) and the European Atherosclerosis Society (EAS) [2,3], the nutritional and dietary approach, along with adopting healthier lifestyles, are the first-line treatments of hypercholesterolemia in children and adolescents. If nutritional and lifestyle treatments prove ineffective in modifying LDL cholesterol, and/or if the hypercholesterolemia is severe, pharmacological intervention must be considered. Adult subjects with dyslipidemia have complemented their diet with dietary supplements or fortified foods for decades, and these compounds have also been recommended for pediatric subjects with familial hypercholesterolemia (FH) in the 2016 EAS guidelines [4]. In this narrative review, we have examined the key consensus documents, meta-analyses, and Randomized Controlled Trials (RCTs) to analyze the main characteristics of nutritional intervention for hypercholesterolemia in childhood. In the second part, our aim has been to better elucidate the lipid-lowering action of nutraceuticals. Nowadays, detailed guidelines are available for both the nutritional treatment of hypercholesterolemia and the use of nutraceuticals in adult subjects, whereas evidence for nutritional intervention in pediatric patients with hypercholesterolemia is more generic and less consistent. Our review aims to fill this gap by analyzing in detail the current evidence on the use of nutraceuticals for pediatric patients with dyslipidemia. Additionally, we discuss the most extensively studied and used nutraceuticals in pediatric patients with hypercholesterolemia. Furthermore, we describe their mechanism of action on cholesterol metabolism, present the main scientific evidence in this field, and summarize the recommendations provided by international guidelines. Finally, we have highlighted the main strengths and limitations/risks of nutraceutical use for lipid-lowering treatment in pediatric-age patients.

## 2. Materials and Methods

The MEDLINE-PubMed database was systematically searched to collect and select publications spanning from 1990 to 2023. The search criteria included randomized placebo-controlled trials, controlled clinical trials, double-blind, randomized controlled studies, and systematic reviews. The following combinations of keywords were used: “nutritional intervention” OR “nutraceuticals” AND “hypercholesterolemia” OR “familial hypercholesterolemia” OR “dyslipidemia” AND “pediatric” OR “children” AND “cardiovascular risk prevention”. We also conducted a manual search of the reference lists of the selected studies. The search was confined to English-language journals, and only full papers were considered.

## 3. Familial Hypercholesterolemia and Atherosclerosis

FH is a genetically determined disease characterized by elevated plasma total and LDL cholesterol levels from the earliest years of life, leading to early atherosclerosis [5]. Genetic defects underlying FH affect key genes involved in LDL cholesterol metabolism and synthesis (Low Density Lipoprotein Receptor (LDL-R), Apolipoprotein B (apoB), Proprotein Convertase Subtilisin/Kexin type 9 (PCSK9) and low-density lipoprotein receptor adaptor protein 1 (LDLRAP1)) [6]. The FH prevalence varies, with 1 in 300 individuals in the heterozygous form (HeFH) and 1 in 360,000 individuals affected by the homozygous form [7]. Prolonged and cumulative exposure to elevated LDL cholesterol plasma levels from early childhood induces vascular modifications and initiates the atherosclerotic process that, in subjects with HoFH, leads to premature coronary atherosclerosis and even myocardial infarction, already at the pediatric age [8]. HeFH is the most common FH form; Clinical symptoms may not manifest in childhood, though, due to its codominant autosomal genetic transmission. However, it is possible to detect one of the parents who has experienced a cardiovascular event, and/or one grandparent who has experienced myocardial infarction or a cardiovascular event at a young age. Early diagnosis, ideally in childhood, is of paramount importance to treat FH as soon as possible, and to reduce CHD-related morbidity and mortality [7]. Over the past two decades, various cholesterol screening strategies have been implemented worldwide, with different characteristics according to each country, leading to relevant improvements in the FH detection rate. However, FH remains highly undetected, resulting in underdiagnosis and subsequent undertreatment [9]. According to prevailing consensus documents available so far [2,4], nutritional and lifestyle interventions constitute the first-line approaches to manage subjects with FH, even in cases of severe hypercholesterolemia, particularly during childhood. For pediatric patients, establishing healthy nutritional habits in the early years of life enhances the likelihood of their persistence throughout the lifespan [3]. 

### 3.1. Nutritional Intervention in Subjects with FH

Nutritional intervention for pediatric patients with dyslipidemia represents a complex and delicate matter, as it necessitates balancing the requirements for growth and neurodevelopment alongside the modulation of plasma lipid profiles through dietary lipid intake [10]. The outcomes of nutritional intervention on plasma lipid values have been explored in several trials [11], focusing on the reduction in saturated and hydrogenated fat intake [12], coupled with a well-balanced intake of carbohydrates and proteins [13]. Data from the Dietary Intervention Study in Children (DISC) [14] and the Special Turku Coronary Risk Factor Intervention Project (STRIP) [15] studies have affirmed the safety of nutritional interventions aimed at lipid reduction in childhood. These results have contributed to the current recommendations outlined in the main consensus documents [3,4].

#### 3.1.1. Lipid Restriction

Quantitative and qualitative restrictions on lipid intake constitute the primary nutritional intervention in the treatment of dyslipidemias [16]. Overall, energy requirements are higher in childhood compared to in adult subjects to meet growth and neurodevelopment needs, and lipids are fundamental components of many organs and systems, such as the central nervous system and retina. To meet these higher needs, infants’ and toddlers’ nutrition typically requires a high lipid intake (35–45% of total daily energy intake) [17], as provided by human milk, which remains the gold standard food, even for infants with severe hypercholesterolemia [16]. Human milk, indeed, provides more cholesterol than any formula milk, but it has been demonstrated that it serves as a positive epigenetic factor, playing a preventive and protective role against the development of hypercholesterolemia [18], blood hypertension [19], obesity [20], and eating disorders (both in terms of quantity of foods and on preference for healthier foods) [21,22] later in life. Complementary feeding (formerly defined as weaning) corresponds to the introduction of solid foods, typically offering lower lipid and higher carbohydrate and protein contents when neither human nor formula milk intake can sufficiently meet the macro and micronutrients needs of infants. This crucial shift in infants’ nutrition is strongly influenced by family habits, culture, and traditions, which vary in each country [23]. Studies on complementary feeding are often conducted on small samples and can rarely be compared [24]. Beyond two years of age, the lipid intake should be reduced to 30% of the total daily energy, and lipid-rich foods should be replaced by cereals, fruits, vegetables, dairy products with low lipid content, pulses, lean meats, and fish [25]. During these years, it is crucial for pediatricians to promote a qualitatively adequate diet, avoiding caloric excess, even by recommending low-fat (partially or totally skimmed) dairy products [26]. This preventive nutritional intervention may be very useful to counteract the development of nutritional unhealthy habits, which become very common during adolescence. Indeed, adolescents often eat outside, making self-determined food choices; therefore, nutritional mistakes such as skipping breakfast, disordinate meals, and the excessive consumption of junk food and sweetened beverages may become very common, posing a high risk of malnutrition. To maintain an adequate and balanced diet, the subject’s active involvement in any nutritional intervention is of the utmost importance in this age group [16].

#### 3.1.2. Carbohydrates and Proteins

Carbohydrates and proteins are macronutrients that usually have less influence on the total and LDL cholesterol plasma levels. However, they are a fundamental part of a balanced diet, which should be adapted to the needs of children and adolescents [27]. Protein intake seems to be quite stable at different ages [17]; nevertheless, excessive protein consumption has been demonstrated to be linked to the development of obesity [28,29] at any age. Starting from complementary feeding, carbohydrates become the main energy source in children’s diet, and according to the main consensus documents [30], they should account for 45 to 60% of the total daily energy intake. Carbohydrates primarily influence triglycerides and High-Density Lipoprotein (HDL) cholesterol plasma levels [31]. Specifically, simple sugars should constitute less than 10% of the total daily energy intake, not only to limit triglyceride plasma levels but also to reduce the risk of developing metabolic syndrome [32,33]. The intake of complex carbohydrates (whole foods) and soluble fibers (as found in pulses, fruit, vegetables, and whole cereals) represents a healthy energy source that can effectively balance a lower lipid intake [34]. According to the 2019 EAS/ESC Guidelines, subjects on a low-lipid diet are recommended to have a daily fiber intake of 25–40 g, with 7–13% being soluble fibers; for pediatric patients, the recommended daily fiber intake is 8.4 g/1000 kcal [17,30].

### 3.2. Ideal Diet and Dietary Patterns

Nowadays, attention is primarily focused not only on macronutrient intake, but also on the so-called “dietary patterns”, referring to the combination of different foods and nutrients to set up a healthy diet. Subjects with FH should follow “healthy heart dietary patterns”, and the Mediterranean Diet (MD) represents the ideal dietary model for reducing the risk of cardiovascular disease [35]. The MD is characterized by a high intake of vegetables, fruits, pulses, and cereals (often pasta or rice), a low consumption of meat and its derivatives, and an adequate intake of dairy products. Fish should be included in the diet many times a week, and lipid intake is mainly derived from olive oil (unsaturated fats) [36]. Other nutritional models, such as the Nordic Diet (ND) and the DASH Diet, are specifically designed to align with the precise dietary traditions of different countries and geographical areas. All these dietary patterns are characterized by a high consumption of fruits, vegetables, whole cereals, pulses, fish, and nuts, while keeping the intake of meat, dairy products, and simple sugars low [37,38]. In a recent study [39], a nutritional intervention based on two widely adopted dietary patterns (MD and ND) was evaluated in children with FH. Both analyzed dietary patterns led to an equal reduction in the plasma LDL cholesterol levels; consequently, the authors concluded that lipid-lowering nutritional interventions can be designed and adjusted according to local traditions and the cultural heritage of different countries and geographical areas.

## 4. Nutraceuticals in the Treatment of Familial Hypercholesterolemia

The term “nutraceutical”, coined by DeFelice in 1989, originates from the fusion of “nutrition” and “pharmaceutical”. It refers to a food or a component of food that can exert a positive impact on human health, comprising preventive and therapeutic actions for a specific disease. Nutraceuticals mildly lower the lipid plasma levels, and their safety and tolerability profile in adulthood are universally proven. The action of nutraceuticals on lipid metabolism involves various pathways modulating the inhibition of cholesterol absorption, synthesis, and metabolism; this multifaceted approach can be complemented with dietary and lifestyle intervention, other nutritional compounds, and drug treatments [40]. Nutritional compounds favor several pleiotropic outcomes: they improve the activity of the endothelium and the characteristics of arterial vessels’ walls while counteracting inflammation and oxidative processes [41]. Additionally, patients who do not tolerate statin therapy often find nutraceuticals to be a more acceptable alternative [42]. Subjects aged over 75 years and subjects who, despite being treated with statins or ezetimibe, have LDL cholesterol values off target, may achieve a better lipid profile by combining drug therapy with nutraceuticals [43]. In fact, the European Atherosclerosis Society indicates [4] nutraceuticals as compounds that can be used to lower cholesterol plasma values in well-defined groups of patients with hypercholesterolemia (subjects older than 18 years of age and pediatric patients aged more than six years). The effects of nutraceuticals in pediatric patients with dyslipidemia have been analyzed in some randomized controlled protocols, most times involving small population samples. The main studies on the use of nutritional compounds in pediatric patients focus on fibers and phytosterols/stanols. In contrast, studies regarding red yeast, soy proteins, probiotics, omega-3 fatty acids, and nuts are occasional and not aggregate [44]. In the 2021 European Society of Cardiology (ESC) Guidelines on cardiovascular disease prevention in clinical practice [45], nutraceuticals are described as potentially useful in dyslipidemia treatment, yet the lack of evidence-based studies for most nutraceuticals is highlighted. Moreover, a warning is issued regarding the absence of studies stating that nutritional compounds may exert a preventive action against CHD-related morbidity and mortality [44]. Lipid nanoparticles are bioactive carrier systems that can improve transport, pharmacokinetics, and stability of encapsulated nutraceuticals; nanoemulsions and microemulsions have also been employed [46]. These formulations have been utilized to deliver nutraceuticals for the treatment of hypercholesterolemia in adult subjects, such as polyphenols and bergamot [47]. Some, but not all, guidelines for dyslipidemia include nutraceuticals among potentially effective treatment options for mild dyslipidemia. It should always be remembered, however, that the evidence regarding the beneficial effects of nutraceuticals on certain endpoints, such as heart stroke and CHD-related diseases, is often poor. 

As for their action on cholesterol metabolism, nutraceuticals that can lower plasma lipid values may be subdivided into three main categories: inhibitors of intestinal cholesterol absorption, inhibitors of liver cholesterol synthesis, and inducers of cholesterol excretion. Nevertheless, they often act through multiple and sometimes unclear pathways, yet providing an overall protective effect on lipid metabolism and atherosclerotic process rate decrease [40,41].

### 4.1. Nutraceutical Inhibitors of Intestinal Cholesterol Absorption

Fibers are components of plant foods formed by carbohydrates that resist digestion in the gastrointestinal tract. Viscosity is the mechanism through which fibers exert their lipid-lowering effect: Viscous water-soluble fibers act as a gel and bind to bile salts in the intestine, increasing their elimination through feces. Bile is mainly synthesized by cholesterol; thus, when bile salts are excreted at a higher rate, a greater quantity of cholesterol may be available for liver bile synthesis. The higher the fiber’s viscosity, the more extensive their cholesterol-lowering effect [48]. Moreover, short-chain fatty acids (SCFA), produced through gut fiber fermentation, may play a protective role in terms of lipid profile modification [49]. The positive impact of fiber on fat-related metabolic pathways has been proven, as fiber helps to reduce both the total and LDL cholesterol levels in the plasma [50], and this has also been acknowledged by the European Food Safety Authority (EFSA) [51]. The effect of fiber intake on lipid plasma values has been analyzed in various studies [52,53]. Nutritional supplementation with oat β-glucan [54,55], psyllium [56,57], pectin, guar gum, and glucomannan [58] significantly reduces the LDL cholesterol plasma values [59]. The majority of trials investigating the lipid-lowering effects of fibers in pediatric subjects highlighted a satisfying adherence to the given treatment, which may be attributed to the pleasant taste of nutraceuticals. The side effects of diarrhea and abdominal pain were only sporadically reported [44,59]. However, the prolonged supplementation of fibers in children and adolescents cannot yet be considered entirely safe, as data on this topic are still limited. 

Phytosterols and stanols are bioactive components derived from plants, sharing a structural resemblance to cholesterol. Phytosterols are steroid alkaloids similar to cholesterol, except for their lateral chain. In contrast, stanols are 5α-saturated derivatives of plant sterols. Since phytosterols and stanols cannot be synthesized by humans, their dietary intake through food is crucial. Foods rich in these compounds include fresh fruits, nuts, vegetables, seeds, cereals, pulses, and vegetable oils [60]. The evidence regarding the effect of phytosterols in pediatric subjects is still limited, but recent trials have demonstrated that supplementation with plant sterols is associated with a decrease in the total cholesterol plasma values in pediatric subjects with moderate hypercholesterolemia [61] and in those with familial hypercholesterolemia [62]. Supplementing pediatric subjects with familial hypercholesterolemia following the CHILD I or CHILD II dietary treatment [2] with 1.2–2 g/day of plant sterols resulted in an over 10% decrease in plasma LDL cholesterol values. Furthermore, increasing the daily intake of phytosterols to 2.3 g led to an even greater reduction in LDL cholesterol plasma values [62]. Plant sterols and stanols typically exhibit a good safety profile, with reported adverse effects being minor up to now. However, available evidence in this context is still limited, and further studies on longer supplementation periods are necessary [44]. Plant sterols and stanols should be considered as valuable therapeutic options in pediatric patients with hypercholesterolemia. It is important to bear in mind that nutritional and lifestyle counseling, and when necessary, pharmacologic therapy, are the cornerstones of treatment during the developmental age [63].

Probiotics are defined as “vital microorganisms that, when consumed in sufficient quantity, confer health benefits to the host”. Recently, a few studies have reported a positive outcome of probiotics on lipid profiles. However, the trials conducted so far lack homogeneity in terms of their duration, the strains of probiotics, dosage, study population, and the types of carriers [64]. Probiotics influence lipid metabolism, but the molecular mechanisms involved are still not clear and well defined. One hypothesis is that probiotics can bind to cholesterol in the intestine or incorporate it into their cell membrane. Lactobacillus acidophilus and Lactobacillus bulgaricus possess enzymes capable of catalyzing cholesterol modifications, thus exerting a favorable action on cholesterol elimination through feces [64]. Other probiotics can reduce the gut–liver flow of bile salts by activating bile salt hydrolase. Some strains of Lactobacilli and Bifidobacteria act by deconjugating bile acids through an enzyme-like mechanism, enhancing their elimination and promoting the systemic hepatic mobilization of cholesterol for the de novo synthesis of bile salts [65]. Certain probiotics may also influence the gut pH, micelle formation, and the transport of lipoproteins, cholesterol, and cholesterol esters [64]. The mechanisms by which probiotics act on lipid metabolism are currently theoretical, and further evidence is needed to determine the extent of their effect on the plasma lipid levels. Importantly, probiotics can be used safely, with no significant reported side effects [44]. 

### 4.2. Nutraceutical Inhibitors of Liver Cholesterol Synthesis

Red yeast rice (RYR) and policosanols are nutraceuticals whose principal mechanism of action is the inhibition of hepatic cholesterol synthesis.

Certain specific yeasts (*Monascus purpureus*, *M. pilosus*, *M. floridanus*, *M. ruber*) in rice (*Oryza sativa*) can produce RYR through fermentation processes. RYR has a long history of use in China, where it has been employed for centuries to enhance the taste of food [66]. Monacoline K, derived from red rice fermented by *Monascus purpureus*, inhibits the activity of 3-hydroxy-3-methyl-glutaryl-coenzyme A (HMGCoA) reductase in the liver, consequently reducing endogenous cholesterol synthesis. The structure and function of Monacolin K are similar to those of lovastatin [67]. The lipid-lowering effect of RYR is dependent not only on Monacolin K but also on many different kinds of monacolin and other compounds such as phytosterols, fibers, and niacin, which can also have a positive effect on lipid metabolism [66,67]. In line with the EFSA’s guidance in 2011, RYR which contains Monacolin K ≤ 10 mg/die can be used in adult subjects provided that their CHD risk is mild or moderate and that their LDL cholesterol plasma values do not exceed 25% of the therapeutic goals, despite nutritional and lifestyle treatments [68,69]. However, in the last decade, potential safety concerns related to the use of nutraceuticals and/or foods containing RYR-derived Monacolin have been raised. In June 2022, the Commission Regulation of the EU declared that each nutraceutical portion for daily consumption should contain no more than 3 mg of RYR-derived monacolins. Consequently, nutraceuticals with a daily dose of RYR-derived monacolins higher than or equal to 3 mg/day will be banned in Europe, and mandatory mentions and warnings must be present on the labeling of nutraceuticals with RYR-derived monacolins [70]. RYR has not been extensively studied in children and adolescents. Since Monacolin is similar to lovastatin, individuals younger than 18 years of age receiving RYR should be closely monitored, both biochemically and clinically. It is crucial to ensure that RYR is pure and contaminant free [71]. In conclusion, RYR may be considered for the treatment of pediatric patients with hypercholesterolemia, but those undergoing treatment with RYR must remain under continuous medical supervision and undergo regular biochemical and clinical follow-ups [72].

Policosanols (PCSs) are a combination of long-chain alcohols derived from plant waxes, sugar cane, rice bran, and potatoes [73]. In recent years, PCSs have been extensively used as a treatment for dyslipidemia in Cuba [74]. Available evidence on PCSs’ use in pediatric patients with dyslipidemia is still scarce [72], without clear evidence of their efficacy and safety. At present, PCSs are not recommended in subjects younger than 18 years old [75,76]. 

### 4.3. Nutraceutical Inducers of LDL Cholesterol Excretion 

Some nutraceuticals can promote an increase in LDL cholesterol excretion, thus enhancing LDL receptor expression and prolonging its half-life on the hepatocyte surface. The ultimate result of these processes is the lowering of plasma lipid values. Soy and lupins are the most studied foods exerting this action. 

Soy is a bean (Glycine max) which is derived from an Asian plant and it has several notable dietary characteristics. It contains a high percentage of proteins (36–46%), essential amino acids, lipids (18%), soluble carbohydrates (15%), and fibers (15%). Additionally, soy includes important micronutrients such as soy lecithin (0.5%), sterols (0.5%), and tocopherols (0.02%). Soy has been extensively studied for years for its nutritional properties and significant positive health effects, and epidemiologic data suggest the existence of an inverse relationship between soy intake and CHD [77]. In a recent trial, the impact of soy intake on lipid plasma values was investigated in a population of children with familial hypercholesterolemia who were administered soy for 13 weeks. In the intervention group, the LDL cholesterol plasma values were 10% lower after treatment compared to the pre-treatment levels [78]. However, it is worth noting that if soy intake is elevated and prolonged, its isoflavones content could potentially interfere with thyroid function and fertility. In addition, soy has a high content of phytic acid, which may reduce the absorption of calcium, magnesium, copper, iron, and zinc. 

Lupins are legumes with a composition poor in salt and with a low glycemic index, and no phytoestrogens. Lupins’ macronutrients are distributed as follows: proteins (30–35%), fibers (30%), carbohydrates (3–10%), and lipids (6%), with 81% of the lipids being polyunsaturated fatty acids. Lupin use is generally considered safe, and they have only minor and transient adverse effects [79]. However, lupins can lower plasma lipid values based on the consumed dose, which may impact therapy adherence for extended periods [43].

### 4.4. Nutraceuticals with Mixed Action

This category of nutritional compounds includes nutraceuticals with multiple actions, often not totally understood. Polyunsaturated long-chain fatty acids (Lc-PUFAs) are the most analyzed compounds in this category.

Omega-3 Lc-PUFAs can be naturally found in both animals (for example, fish, krill, eggs, squid), and vegetables (for example, seaweeds, nuts, flax seeds, and sage). Protection against cardiovascular disease, as assessed by trials on epidemiology and intervention, have been widely recognized. Recently, the EFSA [80], the American Heart Association (AHA) [81], and the Food Standard of Australia and New Zealand (FSANZ) [82] have endorsed omega-3 LCPUFAs as effective nutraceuticals for CHD. The EFSA states that a 2 g/day intake of docosahexaenoic acid (DHA) and eicosapentaenoic acid (EPA) can help keep plasma triglycerides values within the normal range, whereas the AHA suggests 2–4 g/day DHA and EPA supplementation to reduce triglyceride plasma values by 25–30% [81]. There is limited evidence on the effects of LCPUFAs on lipid profiles in children and adolescents. The European Society for Paediatric Gastroenterology Hepatology and Nutrition (ESPGHAN) has studied their overall healthy effect in pediatric subjects [83], whereas LCPUFAs’ action on lipoproteins in children with dyslipidemia has been described by Engler et al. in the “Effect of Docosahexaenoic Acid on Lipoprotein Subclasses in Hyperlipidemic Children” (EARLY) study [84]. Lc-PUFAs have few adverse effects and their use is considered safe, but they are not always very palatable because they are derived from fish. Seaweed-derived omega-3 Lc-PUFAs should be considered to improve patients’ compliance with therapy. 

The mechanisms of action and main characteristics of the analyzed nutraceuticals are summarized in Table 1.

## 5. Nutraceuticals in Combined Therapy

Scientific advances have led to the development of multiple-activity products (foods or supplements) that are a combination of many bioactive compounds, aiming to achieve an additive lipid-lowering action. Lipid-lowering nutritional compounds can be combined in various ways: RYR and PCS, RYR with polycosanols and berberine, RYR and phytosterols, and so on [85]. These associations have been extensively studied in adulthood, but there are still limited data in pediatric patients, and the current promising available evidence needs further and more robust studies to be confirmed. 

Several trials have also emphasized a further effect of nutraceuticals on drug treatment. The association of nutraceutical with lipid-lowering pharmacological therapy may be useful to reach target plasma lipid values with lower dosages of the pharmacological substances and fewer adverse effects [86]. Nutritional compounds can be used in association with other substances, mainly in those subjects who have a poor tolerance of high-dose statin therapy, and exhibit poor therapeutical compliance [86,87]. However, it is mandatory to stress that these associations have been studied in time-limited trials involving small study populations, and their long-term efficacy is not yet fully convincing. Thus, further studies are needed before adopting these associations for the treatment of pediatric patients.

## 6. Conclusive Considerations

In the first part of this narrative review, we provided an overview of the presently recommended nutritional intervention in pediatric patients with FH. Our emphasis was on delineating the key components and distinctive features of this intervention. In the second part, we delved into the primary nutritional compounds employed in the treatment of pediatric patients with FH. This involved a detailed exploration of their mechanisms of action, accompanied by a compilation of the principal scientifical evidence supporting their efficacy in reducing blood lipid levels.

Indeed, as mentioned earlier, nutraceuticals exhibit remarkable versatility in the management of dyslipidemia by influencing various pathways of lipid metabolism. They achieve a positive impact on lipid profiles trough the concurrent modulation of different metabolic steps (absorption, synthesis, excretion). In clinical practice, nutraceuticals are predominantly perceived as dietary supplements rather than pharmaceuticals. While this perception may enhance the acceptance of the treatment among parents and children, it also introduces challenges. Nutraceuticals are often self-prescribed without direct medical supervision and control, potentially impacting their proper use [44]. Additionally, the cost factor comes into play, as nutraceuticals are frequently more expensive than pharmaceutical alternatives. This financial aspect can affect adherence in terms of both the duration of the consumption and consistency of use.

In Figure 1, we have summarized the main positive and negative aspects of nutraceutical use in pediatric patients with dyslipidemia.

Nutritional compounds that can help decrease cholesterol plasma levels, both as functional foods and supplements, show promise as a therapeutic approach for pediatric subjects. However, it is crucial to consider some potential risks associated with their use. Trials on nutraceuticals have often been conducted on small patient cohorts; therefore, further trials involving multiple centers and larger study populations are needed. Aligning with current scientific evidence and the latest EAS guidelines, we support the use of nutraceuticals containing fibers and phytosterols for pediatric patients with genetically determined dyslipidemia, starting from six years of age [68]. Nutraceuticals containing fibers and plant sterols should be considered as a complement of nutritional and lifestyle treatment, which remain a cornerstone in the management of dyslipidemia in pediatric subjects [88]. In conclusion, until long-term studies validate their safety, nutraceuticals should be used for a delimited duration, and limited to patients intolerant to drug therapy or those unable to take it because they are not old enough [44]. 

## Figures and Tables

**Figure 1 children-11-00250-f001:**
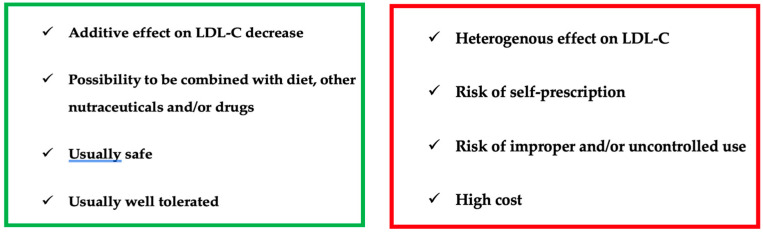
Use of nutraceuticals in children with dyslipidemia: PROS and CONS.

**Table 1 children-11-00250-t001:** Mechanisms of action and main characteristics of nutraceuticals.

Mechanism of Action	Nutraceutical	Main Characteristics
Inhibition of intestinal cholesterol absorption	Fibers	Increase in fecal cholesterol excretion; Inhibition of hepatic cholesterol synthesis.
	Phytosterols and stanols	Decrease in intestinal absorption of exogenous cholesterol; Competition with cholesterol in the formation of solubilized micelles.
	Probiotics	Increase in fecal cholesterol excretion; Inhibition of the formation of solubilized micelles.
Inhibition of liver cholesterol synthesis	Red yeast rice	Inhibition of HMG-CoA reductase, key enzyme in endogenous cholesterol synthesis.
	Policosanols	Reduction in the cellular expression of HMG-CoA reductase, resulting in reduced cholesterol synthesis.
Induction of LDL cholesterol excretion	Soy and lupin proteins	Down-regulation of expression of SREBP-1, with decreased hepatic lipoprotein secretion and cholesterol content; Regulation of SREBP-2, with increased clearance of cholesterol from the blood.
Mixed actions	ω-3 polyunsaturated long-chain fatty acids	Reduced synthesis of hepatic Very-Low-Density Lipoprotein (VLDL); Reduction in available substrate for the synthesis of new triglycerides; Reduction in the activity of triglyceride-synthesizing enzymes.

## Data Availability

All data presented in this article are sourced from previously published articles referenced herein. Specific references to the sources of these data can be found in the reference list. No new data were generated or analyzed for this study.

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
