# Peer review of "Diet and Lipid-Lowering Nutraceuticals in Pediatric Patients with Familial Hypercholesterolemia"

_children, 2024, doi:10.3390/children11020250_

Round 1

Reviewer 1 Report

Comments and Suggestions for Authors

This is an interesting review article with adequate novelty. However, several ponys should be addressed.

- Based on the content of the manuscript, the title should be changed by replacing nutritional compounds with nutraceuticals.

- The above change should be also applied to the abstract. Moreover, the term"nutritional compounds" is very general and there is not a formal definition for them.

- Several abbreviation should be explained when they firstly reported, e.g., LDL, apoB, PCSK, LDLRAP1, HEF, DISC, STRIP, HDL, DASH, HMGCoA, ESPGHAN, VLDL, etc.

- The Introduction section should be split into2 or 3 paragraphe because it is too long.

- In the introduction, the authors should describe the literature gap that their review aim to cover.

- The table are too small and they should be merged into one table.

- The sentence in lines: 69-74 should be split to 2 sentences. Maybe the separation could be in line: 72.

- In the Introduction section, before the aim of the study, the authors should desrcibe the literature gap that exists in the literature for which they performed their review study to cover this literature gap.

- Again in the the section 2 the second paragraphs shouuld be split into two paragraphs. 

- The sentense in lines: 89-93 is quite confusing including also syntax errors. The authors shoul split this sentence into two sentences (e.g, "... present in childhood, due to its codominant autosomal genetic transmission it is possible...".

- A Maerials/and or Methods section is missing. This section is very important for the readers.

- The section 2.1.1 should be split nto 2 oe 3 paragraphs.

- Much more information should be reported concerning the statement in lines 153-154.

- In section 2.2 the sentence in lines 168-173 should be split into two sentences in order to be better understood by the readers.

- The 1st paragrph of section 3 should be split into 2 or 3 paragraph.

- In the section 3.1 at the 1st paragraph, the authors should report additional information.

- Line 217: Please revise as "... Bile is mainly synthesized by cholesterol ..."

Lines 218-219: Please revise as "...  a greater quantity of cholesterol may be available for liver bile synthesis...".

Line 220: Please replace "What is more" with "Moreover".

Line 231, Please  replace "Anyhow" with ""However".

- Line 233, Please revise as "....were derived from ...".

- Line 394, Please revise as "Nutritional ...".

- Line 234: Please revise as "... steroid ...".

- Line 235: Put a dot and start the next sentence as "...On the contrary, stanols ...".

- Line 237: Put a dot and start the next sentence as "... Foods rich in ....".

- Line 246: Plese revise as "adverse".

- Line 253: Please revise as "... but the implicated molecular mechanisms are ...".

- Line 255: Please revise as "... into ...".

- Line 260: Please replease as "... by an ezyme-like ...".

- Line 265: Please revise as "... can be ...".

- Line 273: Please revise as "Specific ...".

- Line 283: Please revise as ".... subjects to reduce CHD risk ...".

- Line 287: Please revise as "... declared...".

- Line 295: "....that it does not contain [71].". Something is missing here.

- Line 295: Please revise as "... can be...".

- Line 296: In phrasing "... but it patients ...". Delete "it".

- Line 342: Please revise as "... cardiovascular...".

- Line 352: Please revise as ".... and they are ".

- The reoslution of the figure should be improved.

- A major issue, concerning the bioactive compounds induding in the nutraceuticals deals with their poor oral bioavailability (e.g., polyphenols, flavonoids, etc.). This issue should be reported into the manuscript.

- Base on the abobe, the authors should include novel technologies that can improve the oral bioavailability of such bioactive compounds (e.g. nanoformulation-nanoparticles, etc.).

- There are several English misspelings, and syntax/grammar errors, as well as typos throughout the manuscript that should be carefully revised.

- The authors should add more references published the last 2-3 years.

-References format is not identical and should be revised according to journal style.

Comments on the Quality of English Language

Extensive editing of English language is highly required.

Author Response

Dear Reviewer #1,

Thank you very much for your insightful comments and for appreciating our manuscript. We have incorporated the following modifications based on your suggestions:

  • Revised the manuscript title.
  • Replaced the term “nutritional compounds” with “nutraceuticals”.
  • Reported all abbreviations.
  • Divided the Introduction section into two paragraphs and added clarifying lines to better convey the aim of our review and the gap it aims to fill.
  • Consolidated the tables into a single table.
  • Added a “Materials and Methods” section.
  • Enhanced the analysis of fiber intake, addressing your suggestion.
  • Included information on nutraceuticals' bioavailability and emerging technologies, such as nano-particles.
  • Updated the “References” section with more recent papers.
  • Carefully reviewed the entire manuscript, incorporating all spelling and language suggestions.
  • Conducted an English language revision.

Best regards.

Reviewer 2 Report

Comments and Suggestions for Authors

Overall, well organized manuscript. narrative was consistent with objectives stated in the introduction.  However, major comment - it is not clear what type of review this is- systematic? narrative?...other? the PRISMA checklist is the standard review protocol. It isn't clarified until page 13 - this needs to be provided to the reader sooner. Would encourage review for grammar / typos. Consider revising title - it is not clear in the title that manuscript is a narrative review. What do authors mean by nutritional compounds? nutraceuticals? please consider consistent terminology for outcomes/variables throughout.

line 204: "instance"  is misspelled.

line 394: "nutritional" is misspelled. 

Comments on the Quality of English Language

n/a

Author Response

Dear Reviewer #2,

Thank you for your valuable comments on our manuscript. We have made the following adjustments based on your suggestions:

  • Revised the manuscript title.
  • Added a 'Materials and Methods' section.
  • Modified the manuscript as per your recommendations.
  • Conducted an English language revision.

Best regards.

Round 2

Reviewer 1 Report

Comments and Suggestions for Authors

The authors significantly improved their manuscript increasing their quality and adequancy.

Comments on the Quality of English Language

Minor editing of English language required